# Dynamic Contrast-Enhanced Ultrasound in the Prediction of Advanced Hepatocellular Carcinoma Response to Systemic and Locoregional Therapies

**DOI:** 10.3390/cancers16030551

**Published:** 2024-01-27

**Authors:** Lucia Cerrito, Maria Elena Ainora, Giuseppe Cuccia, Linda Galasso, Irene Mignini, Giorgio Esposto, Matteo Garcovich, Laura Riccardi, Antonio Gasbarrini, Maria Assunta Zocco

**Affiliations:** 1Department of Internal Medicine and Gastroenterology, Fondazione Policlinico Universitario Agostino Gemelli IRCCS, Catholic University of Rome, 00168 Rome, Italyainoramariaelena@gmail.com (M.E.A.); giuseppe.cuccia@guest.policlinicogemelli.it (G.C.); giorgio.esposto2@gmail.com (G.E.); matteo.garcovich@policlinicogemelli.it (M.G.); laura.riccardi@policlinicogemelli.it (L.R.); antonio.gasbarrini@policlinicogemelli.it (A.G.); 2CEMAD Digestive Disease Center, Fondazione Policlinico Universitario Agostino Gemelli IRCCS, Catholic University of Rome, 00168 Rome, Italy; linda.galasso0817@gmail.com (L.G.); irene.mignini@gmail.com (I.M.)

**Keywords:** hepatocellular carcinoma, systemic therapy, dynamic contrast-enhanced ultrasound, time–intensity curves

## Abstract

**Simple Summary:**

Hepatocellular carcinoma (HCC) is sometimes diagnosed at an advanced stage, with subsequent complex therapeutic efforts combining both locoregional and systemic treatments. Computed tomography and magnetic resonance are conventionally used for post-treatment follow-up of HCC. Contrast-enhanced ultrasound and dynamic contrast-enhanced ultrasound (DCE-US) have gained an increasing importance due to the interest of several researchers in their potential role in the early assessment of response to locoregional treatments or antiangiogenic therapies in patients with advanced HCC. Particularly, DCE-US allows the construction of time–intensity curves, providing an assessment of the parameters related to neoplastic tissue perfusion and its potential changes following therapies. Its advantage resides in being easily repeatable, minimally invasive, and able to grant important evaluations about patients’ survival, essential for well-timed therapeutic changes in case of unsatisfying response and eventual further treatment planning.

**Abstract:**

Hepatocellular carcinoma (HCC) is the most frequent primary liver cancer and the sixth most common malignant tumor in the world, with an incidence of 2–8% per year in patients with hepatic cirrhosis or chronic hepatitis. Despite surveillance schedules, it is sometimes diagnosed at an advanced stage, requiring complex therapeutic efforts with both locoregional and systemic treatments. Traditional radiological tools (computed tomography and magnetic resonance) are used for the post-treatment follow-up of HCC. The first follow-up imaging is performed at 4 weeks after resection or locoregional treatments, or after 3 months from the beginning of systemic therapies, and subsequently every 3 months for the first 2 years. For this reason, these radiological methods do not grant the possibility of an early distinction between good and poor therapeutic response. Contrast-enhanced ultrasound (CEUS) and dynamic contrast-enhanced ultrasound (DCE-US) have gained the interest of several researchers for their potential role in the early assessment of response to locoregional treatments (chemoembolization) or antiangiogenic therapies in patients with advanced HCC. In fact, DCE-US, through a quantitative analysis performed by specific software, allows the construction of time–intensity curves, providing an evaluation of the parameters related to neoplastic tissue perfusion and its potential changes following therapies. It has the invaluable advantage of being easily repeatable, minimally invasive, and able to grant important evaluations regarding patients’ survival, essential for well-timed therapeutic changes in case of unsatisfying response, and eventual further treatment planning.

## 1. Introduction

According to data from the 2020 Global Cancer Observatory, primary liver cancer is the sixth most newly diagnosed cancer worldwide and the third leading cause of cancer death.

Hepatocellular carcinoma (HCC) represents the most frequent primary liver cancer [1]. It has an incidence of 2–8% per year in patients with hepatic cirrhosis or chronic hepatitis, with a strong male preponderance (male/female ratio 2–2.5:1). The main risk factors are hepatitis B virus (HBV) or hepatitis C virus (HCV) infection, chronic alcohol intake, and aflatoxin exposure. Metabolic dysfunction-associated fatty liver disease (MAFLD) plays a progressively increasing role in liver cirrhosis and HCC etiology, and it is expected to become the main HCC cause in the near future [2,3].

The current Clinical Practice Guidelines by the European Association for the Study of the Liver (EASL) endorse the Barcelona Clinic Liver Cancer (BCLC) classification as a HCC staging method, based on tumor-related characteristics, health status (Eastern Cooperative Oncology Group Performance Status—ECOG PS), and liver function. Despite surveillance schedules, HCC is sometimes diagnosed at an advanced stage (BCLC-C), characterized by vascular invasion or extrahepatic diffusion, and requires complex therapeutic efforts with both locoregional and systemic treatments. In these cases, the first-line treatment is represented by the immunotherapeutic agents atezolizumab/bevacizumab, with tyrosine-kinase inhibitors (TKIs) supplied if the previous agents are contraindicated or as further lines of treatment in case of disease progression [4].

## 2. Evaluating the Response of Hepatocellular Carcinoma to Treatments: A Multifaceted and Evolving Landscape

Post-treatment follow-up is usually based on traditional radiological tools such as computed tomography (CT) and magnetic resonance imaging (MRI). The first follow-up imaging is performed 4 weeks after liver resection or locoregional treatments and 3 months after the beginning of systemic therapies. Subsequently, imaging is conducted every 3 months for the first 2 years [5]. According to EASL recommendations, the evaluation of treatment response is assessed according to the modified Response Evaluation Criteria In Solid Tumors (mRECIST). In patients with advanced HCC, a more specific assessment of systemic treatment response can be granted by the Response Evaluation Criteria in Cancer of the Liver (RECICL) that describes the necrotic areas more accurately and defines the concept of progression in a different way [6,7,8].

Contrast-enhanced ultrasonography (CEUS) is increasingly emerging as a valuable method to evaluate both locoregional and systemic treatment responses, as already demonstrated in several studies with attractive results [9,10]. It has been shown that early changes in tumor perfusion patterns could predict long-term response to treatment in different stages of HCC [11]. Other studies hypothesized a role of CEUS also for post-treatment follow-up. In particular, Bansal et al. demonstrated similar performances of CEUS and MRI in HCC recurrence detection and also the ability of CEUS to characterize lesions that were indefinite at MRI [12].

In patients with advanced HCC, both atezolizumab/bevacizumab and TKIs produce modifications in the vascularization of the neoplastic tissue, leading to necrosis. For these reasons, the ideal method for radiological disease assessment should be able to measure early response biomarkers (e.g., changes in neoplastic tissue perfusion and amount of intranodular necrotic tissue) more than the HCC dimensional modifications that occur later. This approach could be at the foundation of a personalized treatment that allows either therapy confirmation, dosage increase, or a change in the therapeutic line.

CEUS is a versatile ultrasound technique performed with intravenous administration of gas-filled microbubbles. Ultrasound contrast media, unlike those used in CT or MRI, remain confined to the vascular space [13] and allow a dynamic study of a liver lesion in real-time, with high temporal and spatial resolution [14]. Current guidelines recommend CEUS for the discrimination of benign and malignant liver lesions or in the follow-up of patients with HCC after locoregional treatments. In recent years, this technique has gained growing importance in the control of vascular modifications of tumor tissue during treatments, but large clinical trials are still needed to completely confirm their role in this field [11,15,16,17].

More recently, dynamic contrast-enhanced ultrasound (DCE-US) has been introduced to overcome CEUS subjectivity; through a quantitative analysis performed by specific software, DCE-US grants quantitative information on contrast media kinetics [18], allowing the construction of time–intensity curves that provide an objective evaluation of the parameters related to neoplastic tissue perfusion and their potential changes after therapy [19,20].

Compared with CT and MRI, this technique can detect signs of response much earlier than traditional radiological tools (even 48 h after treatment), significantly optimizing the time of action in treating neoplastic lesions, with accuracy ranging from 84% to 92% [11,21,22,23].

Currently, the follow-up of treated neoplastic lesions is based on mRECIST criteria, which take into account not only morphological changes but also vascular response in the target lesions [6,24]. DCE-US is a valuable alternative for the analysis of the HCC microvascular district during systemic treatments [25,26,27,28]: it has already been applied in the evaluation of therapeutic response to anti-angiogenic drugs in gastrointestinal stromal tumors (GIST) [29], renal carcinoma [30,31,32], and colon carcinomas [33,34].

DCE-US has gained a rising interest also in advanced HCC for its potential role in the early assessment of response to locoregional treatments or antiangiogenic therapies. Several studies compared the efficacy of this technique versus perfusion CT in assessing the response to systemic sorafenib therapy [27,35,36,37,38].

This recently developed ultrasound (US) technique involves a first practical phase of raw data retrieval, during which CEUS examination is performed in real time, and a second phase of data processing with mathematical analysis by specific software that allows quantitative measures of blood flow and blood volume parameters. This is performed on uncompressed linear data (raw data), which is linearly proportional to the of microbubble concentration. The results are expressed as intensity values after calculating the arithmetic mean of pixel intensities (Figure 1 and Figure 2). Different perfusion parameters able to characterize both blood volume and blood flow can be extracted from time–intensity curves, in particular peak enhancement (PE), the area under the curve (AUC), wash-in rate (WiR), rise time (RT), time to peak (TTP), and mean transit time (mTT).

Thus, DCE-US grants quantitative measures of blood flow and blood volume parameters within the neoplastic lesion [18,37,39,40].

Finally, among the new technologies for the study of neoplastic lesions, three-dimensional CEUS has gained attention in recent years because it can compensate for the limitations related to 2-dimensional images and grants a higher accuracy in the evaluation of neoplastic vascularization [41,42,43].

## 3. Dynamic Contrast-Enhanced Ultrasound in the Analysis of Advanced Hepatocellular Carcinoma Response to Systemic Treatments

Since the first appearance of antiangiogenetic treatments for HCC, the necessity has emerged for an easy method for the early assessment of response to therapy, faster than with the well-established CT and MRI.

Table 1 summarizes the results of the most relevant studies performed with DCE-US for the evaluation of treatment response.

The 2012 pilot study by Shiozawa was probably one of the first papers suggesting an important role of D-CEUS in the assessment of early response to sorafenib. They examined 14 patients with advanced HCC receiving sorafenib for at least 4 weeks and compared the arrival time parametric imaging using CEUS (at 2 and 4 weeks after treatment) with dynamic CT scan (at 4–8 weeks) in the evaluation of the early response of HCC to sorafenib [38]. In color-mapping, frames of the target lesion at 2 and 4 weeks presented a delay in the arrival time of the contrast agent in 8/14 patients, suggesting a post-therapy rearrangement in HCC vascularization. Significant discrepancies were highlighted between patients with partial response or stable disease (*p* = 0.019) and those with neoplastic progression (*p* = 0.028).

On the preclinical side, Zhu et al. investigated through DCE-US the effects of pazopanib (a pan-vascular endothelial growth factor receptor inhibitor) on human HCC and endothelial cell lines both in vitro and in vivo, aiming to quantify the effects of antiangiogenetic treatments in HCC [45]. They studied, with CEUS, the intratumoral blood perfusion in a subcutaneous model “HCCLM3” and subsequently obtained time–intensity curves that allowed for quantitative analysis compared to histological features (hypoxia index and necrosis). No significant results were achieved with pazopanib on HCC cell lines in vitro in terms of proliferation, although the processes of migration and invasion were inhibited, and apoptosis was induced in two HCC cell lines (PLC/PRF/5 and HCCLM3). A dose-dependent mechanism was observed in the inhibition of proliferation, tubule formation, and migration in human endothelial cells. The inhibition of in vivo neoplastic proliferation by pazopanib was noteworthy in HepG2, HCCLM3, and xenograft models. Several variations were observed in HCC perfusional parameters, assessed with quantitative CEUS. An interesting element is represented by the modifications in the intensity of the signal, due to treatment efficacy, which take place in treated HCCs, even before the effective dimensional reduction. In fact, after 3 weeks of treatment with pazopanib, a reduction in the peak and total perfusion of the neoplastic lesion and an increase in mTT of contrast medium were detected. Moreover, Zhu et al. noticed that the density of intratumoral microvessels was inversely related to the mTT of contrast medium in HCC hotspot areas (*p* = 0.001) in the treated group (*p* = 0.039), but not in the controls (*p* = 0.332).

In 2013, when sorafenib still represented the only systemic therapy available for HCC, a study from our group underlined the importance of DCE-US as an early predictor of response to sorafenib and as an efficient instrument to identify the patients who could benefit from anti-angiogenetic therapy [35]. Twenty-eight patients with HCC undergoing sorafenib treatment (400 mg bid) were included in a prospective study and studied with DCE-US at baseline, and after 15 and 30 days of treatment. A comparison was performed between sorafenib responders/non-responders, analyzing the variations in five DCE-US parameters: PE, TTP, slope coefficiency of wash-in (Pw), AUC, and mTT. Significant modifications in tumor vascularization were quantified by the decrease in PE (*p* < 0.001), Pw (*p* = 0.003), and AUC (*p* = 0.002) in sorafenib responders at both 15 and 30 days; non-responders had no relevant variations in time–intensity curves due to the persistence of intralesional perfusional signals. Interestingly, mean overall survival (OS) was higher in responders than in non-responders (382 versus 158 days; *p* = 0.003), whereas a noteworthy correlation was identified between progression-free survival (PFS) and Pw, Tp, and AUC.

Knieling et al. reported a case of a patient with metastatic HCC undergoing treatment with a combination of sorafenib and panobinostat (LBH-589, a histone deacetylase inhibitor) [37]. The outcome during the treatment was monitored through DCE-US and MRI, which contemporarily detected the presence of necrotic areas, demonstrating an excellent correlation between the two imaging methods. Using DCE-US analysis, the authors detected a relevant increase in mTT (baseline: 11.04 s; after 3 months: 17.48 s; and after 5 months: 26.60 s), and TTP (baseline: 8.83 s; after 3 months: 12.32 s; and after 5 months: 15.25 s) relatively early during the treatment, while significant modifications were detected after at least 5 months using MRI. The authors suggested DCE-US as a more sensitive method to detect early changes in HCC microcirculation, providing crucial, specific data on treatment trends, with subsequent therapeutic implications even earlier than the first month of treatment.

Another study by the same group compared the DCE-US parameters evaluated at specific time points (baseline, 1 month, and 3 months) in two groups of patients with advanced HCC treated with sorafenib or transarterial chemoembolization (TACE) [43]. During HCC treatment, the authors observed modifications in blood flow parameters, especially in TTP, which showed a significant increase in responders earlier than one month from the first sorafenib administration. These data foreshadow the suitability of DCE-US for the assessment of HCC response to sorafenib.

The potential role of DCE-US as a tool for the early measurement of tumor response has been evaluated also in a prospective open-label phase II trial by Lo et al., performed on 15 patients with advanced HCC undergoing treatment with axitinib, a selective tyrosine kinase inhibitor (TKI) [26]. The authors observed a decrease in fractional blood volume 2 weeks after axitinib infusion in 10 patients and an increase in 5 patients, with a borderline statistically significant relation to OS (*p* = 0.050). On the other hand, no modifications were detected in the target lesion size, and no significant association was found with PFS (*p* = 0.310), probably due to the small sample size of the study and to other concomitant factors potentially interfering with the clinical endpoints (for example, liver failure relater to underlying cirrhosis).

More relevant are the results of Lassau et al. in patients with advanced HCC treated with bevacizumab, a humanized monoclonal antibody binding vascular endothelial growth factor (VEGF)-A [25]. In this study, DCE-US was performed to quantify the early modifications in neoplastic vascularization as early as three days after bevacizumab infusion and their role as potential predictors of tumor response, OS, and PFS. The decrease in some parameters related to blood volume (AUC and Pw) after three days of treatment showed a trend towards correlation with RECIST response after two months. Similarly, in the survival analysis, the decrease in AUC and AUC during wash-out was associated with OS, whereas the decrease in Pw was associated with PFS.

Both DCE-US and perfusion CT performed one month after the beginning of therapy were able to identify perfusion changes in patients with advanced HCC treated with sorafenib or sunitinib. However, only DCE-US predicted non-progression at 2 months; a decrease in AUC > 60% was associated with treatment response [27].

The majority of these studies involved a single center (taking under examination a limited number of patients) and demonstrated the association of AUC, a parameter related to blood volume, with the response of HCC to treatment with sorafenib or bevacizumab in terms of RECIST criteria [25,35]. An important upgrade in this field was achieved in 2014 by Lassau et al., who conducted a multicentric study aiming to validate the role of DCE-US in the definition of the response to antiangiogenetic treatments in 539 patients with different solid tumors (including 107 with HCC). The comparison between DCE-US at baseline and after 30 days detected several modifications in DCE-US parameters, but the most significant was undoubtedly the association between AUC and freedom from progression (FFP; *p* = 0.00002): a decrease in AUC superior to 40% (after 30 days from baseline) was correlated to better results in terms of both OS (*p* = 0.05) and FFP (*p* = 0.005) [28]. Another relevant analysis was contemporarily performed by the authors in terms of pharmacoeconomy through the assessment of costs. The mean price of each DCE-US was EUR 180–278 (about USD 250–381), with a relevant advantage compared to traditional radiological tools. The authors demonstrated that, due to the possibility of providing data about tumor progression and the economic affordability of DCE-US, this method should be included in the routine assessment of neoplastic lesions for the prediction of OS and tumor progression during antiangiogenetic treatments [28].

Finally, DCE-US parameters were able to predict not only treatment response but also the occurrence of adverse events, as demonstrated by Sugimoto et al. in 37 patients with intermediate and advanced HCC treated with sorafenib. Perfusion parameters related to AUC granted more satisfying results in terms of statistical significance; AUC during wash-in on day 14 was associated with tumor response (*p* = 0.0016), and the ratio between days 0 and 7 was associated with OS (*p* = 0.037) and PFS (*p* = 0.009). On the other hand, a decrease in total AUC of the liver tissue between days 0 and 7 was associated with major drug toxicity (*p* = 0.0002) [46].

Recently, Takada et al. [47] evaluated the role of DCE-US in the prediction of the response to atezolizumab/bevacizumab in a group of 35 patients with advanced HCC. CEUS was performed before treatment and 3–7 days after the first administration. Standard disease assessment was granted by CT or MRI at 8–12 weeks after the beginning of therapy. The authors observed that the absence of a decrease in blood flow parameters was associated with disease progression (60 vs. 0%; *p* = 0.001) and reduced PFS (9.1 vs. 28 weeks; *p* = 0.0051).

## 4. Dynamic Contrast-Enhanced Ultrasound in the Analysis of Advanced Hepatocellular Carcinoma Response to Intraarterial Treatments

Patients with non-resectable/advanced HCC could also be treated with TACE, frequently as a propaedeutic or complementary method to systemic therapies. Even in these situations, DCE-US could become an invaluable resource in the early assessment of tumor response to treatment.

Sparchez et al. underlined the role of CEUS in the early evaluation (7 days) of disease burden after TACE and in the prediction of the need for further treatments. They suggested a possible superiority of CEUS over CT [48].

Similar results were obtained by Uller et al. in a group of HCC patients treated with drug-eluting bead TACE (DEB-TACE). They performed DCE-US before and after treatment and hypothesized a role for the technique as a peri-interventional instrument, useful for the identification of extrahepatic tumor-feeding arteries and for the early evaluation of treatment response [49].

Moschouris et al. confirmed the role of CEUS in the evaluation of treatment response after TACE and demonstrated an elevated concordance with traditional radiological imaging. They also demonstrated a correlation between tumor response after TACE assessed using CEUS and clinical outcomes: responders had longer OS (37.1 vs. 11.0 months; *p* < 0.001) and PFS (24.6 vs. 10.9 months; *p* = 0.007) [50].

A feasibility study by Wiggermann et al. applied DCE-US serial measurements in the evaluation of microcirculation changes taking place in six patients with advanced HCC treated with degradable starch microsphere (DMS)-TACE. The authors quantified the post-TACE perfusion of HCC lesions and identified a reduction in microvascularization in all patients immediately after the procedure. They showed a reduction in PE (9.4 ± 9.1 vs. 34.3 ± 13.1; *p* < 0.001), regional blood flow (4.8 ± 3.4 vs. 34.7 ± 13.4; *p* < 0.001), and regional blood volume (70.9 ± 23.8 vs. 446.5 ± 122.4; *p* < 0.001) compared to pre-TACE evaluation. Due to the transient occlusion of small arterial vessels, gradual revascularization was observed within 120 min from treatment and was characterized by perfusional parameters similar to pre-TACE ones. The authors associated this event with the reduction in peri-procedural washout of the cytostatic agent, granting its temporary storage in the targeted lesion with less systemic diffusion and lower toxicity [51].

A prospective study by Cao et al. analyzed 40 patients with advanced HCC undergoing dynamic three-dimensional (3D) CEUS before and 1–3 days after TACE in order to evaluate microperfusional changes [52]. Quantitative parameters derived from DCE-US were compared between responders and non-responders. Despite the absence of differences in lesion diameters, the authors found a significant reduction in AUC (*p* = 0.047), AUC during wash out (*p* = 0.049), and PE (*p* = 0.022) in responders compared to non-responders. No significant differences were observed in time-related parameters (TTP and mTT). Based on these results, 3D-CEUS could gain more importance in the future, replacing dynamic two-dimensional (2D)-CEUS for early quantitative assessment of microvascularization changes after TACE.

Similarly, Nam et al. compared dynamic 2D CEUS and 3D CEUS performed at baseline, 1–2 weeks after TACE, and about 1 month after TACE in advanced HCC [43]. At baseline (before the procedure), PE from 3D-CEUS was similar in responders and non-responders, but it demonstrated a more pronounced decrease during follow-up in the first group of patients. On the other hand, 2D-CEUS was less efficient since PE was significantly different only one month after the procedure. Moreover, the authors demonstrated a good agreement between 2D-CEUS and 3D-CEUS performed 1–2 weeks and 1 month after TACE, and MRI at 1 month.

Table 2 summarizes the results of the most relevant studies performed with DCE-US regarding response to intraarterial treatment.

## 5. Conclusions

The increasing interest in DCE-US foreshadows its potential role as a crucial diagnostic and prognostic instrument in the definition of advanced HCC response to systemic and locoregional treatment. Parameters derived from time–intensity curves allow a quantitative evaluation of tumor microvascularization, intercepting its modification earlier than traditional radiological methods.

DCE-US has the invaluable advantage of being easily repeatable, minimally invasive, non-nephrotoxic, cost-effective, and able to grant important evaluations regarding patients’ survival. All these features are essential for well-timed therapeutic changes in case of an unsatisfying response and eventual further treatment planning. In fact, an undeniable advantage of the early detection of HCC responders to systemic therapies resides in the possibility to create a personalized therapeutic scheme, potentially saving healthcare resources, improving patients’ prognosis, and decreasing side effects.

Of course, this technique suffers the same limitations as traditional US related to HCC dimensions (large nodules could have a heterogeneous US pattern), lesion locations (CEUS performance is difficult for deep nodules), patient’s habitus or poor compliance, meteorism, and respiratory movements. However, the above-mentioned studies demonstrated the promising role of DCE-US in the strict follow-up of patients with advanced HCC.

Future clinical trials with larger sample sizes are needed to confirm the clinical application of DCE-US in monitoring treatment response and for a possible future introduction of this method in the official follow-up algorithm of HCC undergoing treatment. The design of possible future trials should consider that DCE-US could have a relevant role in the assessment of tumor prognosis and in the identification of tumor progression earlier than traditional radiological imaging.

## Figures and Tables

**Figure 1 cancers-16-00551-f001:**
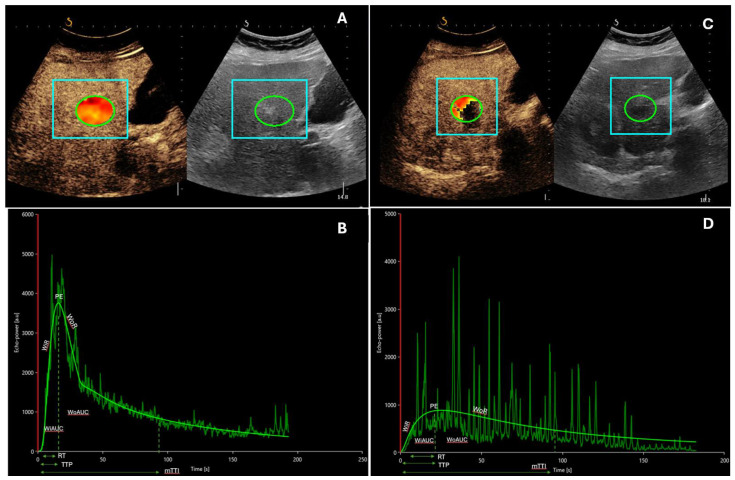
A 59-year-old patient with alcohol-related liver cirrhosis and HCC nodule of 25 mm in the segment V-VI undergoing locoregional treatment with radiofrequency ablation. (**A**,**B**) Contrast-enhanced ultrasound with corresponding time–intensity curves of the lesion before the treatment. (**C**,**D**) Contrast-enhanced ultrasound with corresponding time–intensity curves of the lesion after the treatment. It is possible to observe lower values of PE, AUC, and WiR (expression of blood volume and flow) compared to baseline values. In figures A and C it is possible to identify the regions of interest (ROI) in liver parenchyma (square boxes) and nodule of HCC (circles) Abbreviations: mTT, mean transit time; PE, peak enhancement; RT, rise time; TTP, time to peak; WiAUC, wash-in area under the curve; WiR, wash-in rate; WoAUC, wash-out area under the curve; WoR, wash-out rate.

**Figure 2 cancers-16-00551-f002:**
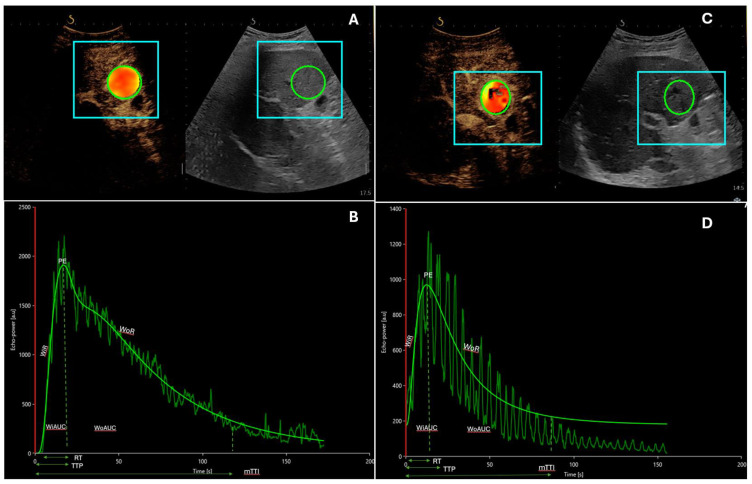
An 82-year-old patient with multinodular HCC developed on steatotic liver disease, undergoing systemic treatment with immunotherapy every 21 days (Atezolizumab 1200 mg and Bevacizumab 15 mg/kg). (**A**,**B**) Contrast-enhanced ultrasound with corresponding time–intensity curves of the lesion before the first cycle of atezolizumab/bevacizumab. (**C**,**D**) Contrast-enhanced ultrasound with corresponding time–intensity curves of the lesion after four cycles of systemic treatment. In this case, post-treatment quantitative parameters related to blood volume and flow are substantially similar to baseline values, demonstrating poor response to therapy. In figures A and C it is possible to identify the regions of interest (ROI) in liver parenchyma (square boxes) and in HCC nodule (circles) Abbreviations: mTT, mean transit time; PE, peak enhancement; RT, rise time; TTP, time to peak; WiAUC, wash-in area under the curve; WiR, wash-in rate; WoAUC, wash-out area under the curve; WoR, wash-out rate.

**Table 1 cancers-16-00551-t001:** Dynamic contrast-enhanced ultrasound (D-CEUS) application in the monitoring of advanced hepatocellular carcinoma (HCC) response to systemic treatments.

Author	Patients (n)	US Contrast Agent	Therapy	
Frampas [27]	16 3	SonoVue^®^, Bracco, Italy	Sorafenib Sunitinib	Opposite modifications in non-progressors and progressors: mean AUC −38.3 vs. 436.3% (*p* = 0.06); mean AUCWI −37.5 vs. 1107.3% (*p* = 0.13); mean AUCWO −37.9 vs. 377.8% (*p* = 0.05). D-CEUS at one month: non-progressors: decrease in AUC > 40%; progressors: decrease in AUC < 40%; *p* = 0.015.
Knieling [43]	9 10	SonoVue^®^, Bracco, Italy	Sorafenib TACE	Increase: Pw → baseline 11.28 s ± 2.03 s (1.00); after 1 month 13.60 s ± 1.52 s (1.53 ± 0.08; *p* = 0.0405); after 3 months 16.17 s ± 2.35 s (1.46 ± 0.07; *p* = 0.0071).
Knieling [36]	1	NA	Sorafenib	Increase: mTT → baseline: 11.04 s; after 3 months: 17.48 s; after 5 months: 26.60 s); Pw → baseline: 8.83 s; after 3 months: 12.32 s; after 5 months: 15.25 s).
Lassau [25]	42	SonoVue^®^, Bracco, Italy	Bevacizumab	Pw reduction at day 3 revealed a trend of correspondence with PFS (*p* = 0.028)
Lassau [28]	539 (107 HCC)	SonoVue^®^, Bracco, Italy	Bevacizumab (for HCC)	Changes in AUC (baseline → day 30): related to freedom from progression (*p* = 0.00002)
Lo [26]	15	Definity^®^; Lantheus, USA (perflutren lipid microspheres)	Axitinib	Median OS: 7.1 months (1.8–27.3 mo; 95% confidence interval [CI]: 0, 14.270)—(*p* = 0.050) Median PFS: 3.6 months (1.8–17.4 mo; 95% CI: 2.085, 5.115)—(*p* = 0.310) No significant association with DCE-US quantitative parameters
Sugimoto [44]	37 16: intermediate HCC 21: advanced HCC	Sonazoid^®^; Daiichi-Sankyo, Japan	Sorafenib	Differences responders versus non-responder (from baseline to day-14): AUC: 0.66 [0.26, 0.82] vs. 1.33 [1.11, 3.65], *p* = 0.0095; AUCWI: 0.64 [0.27, 0.87] vs. 1.81 [1.11, 3.23], *p* = 0.0016; AUCWO: (0.68 [0.26, 0.86] vs. 1.26 [0.62, 3.58], *p* = 0.0177; PI: 0.68 [0.34, 0.88] vs. 1.62 [1.02, 2.26], *p* = 0.0211. Differences in responders versus non-responders (from baseline to day 28): AUCWI: 0.51 [0.32, 0.89] vs. 1.55 [1.17, 2.75], *p* = 0.0222; PI: 0.79 [0.39, 1.00] vs. 1.13 [0.66, 2.32], *p* = 0.1294.
Zocco [34]	28	SonoVue^®^, Bracco, Italy	Sorafenib	Mean overall survival (OS): responders > non-responders (382 versus 158 days; *p* = 0.003) Decrease at 15 and 30 days: peak-intensity (PI; *p* < 0.001), time to PI (Pw; *p* = 0.003), area under the curve (AUC; *p* = 0.002) Correlation between performance free survival (PFS), Pw, Tp, AUC

Abbreviations: area under the curve (AUC); area under the curve wash-in (AUCWI); area under the curve wash-out (AUCWO); dynamic contrast-enhanced ultrasound (D-CEUS); mean transit time (mTT); not available (NA); overall survival (OS); performance free survival (PFS); peak intensity (PI); slope coefficiency of wash-in (Pw).

**Table 2 cancers-16-00551-t002:** Dynamic contrast-enhanced ultrasound (D-CEUS) application in monitoring the response of advanced hepatocellular carcinoma (HCC) to intraarterial treatments.

Study	Method	Findings
Sparchez [47]	DCE-US	Role in early assessment of tumor response to TACE. Early evaluation of disease burden after TACE. Possible superiority over CT.
Uller [48]	DCE-US (before and after DEB-TACE)	Peri-interventional instrument for the identification of extrahepatic tumor-feeding arteries. Early evaluation of treatment response.
Moschouris [49]	CEUS	Evaluation of treatment response after TACE. Elevated concordance with traditional imaging. Correlation with clinical outcomes.
Wiggermann [50]	DCE-US	Applied in a feasibility study for microcirculation changes after DSM-TACE. Reduction in post-TACE perfusion, regional blood flow, and blood volume. Transient occlusion leading to temporary storage of cytostatic agent in the targeted lesion.
Cao [51]	3D CEUS	Analyzed microperfusional changes before and after TACE. Significant reduction in AUC, AUC during washout, and PE in responders. Potential future importance in early quantitative assessment of microvascularization changes after TACE.
Nam [42]	2D CEUS 3D CEUS	Compared dynamic 2D and 3D CEUS at different time points after TACE. The 3D-CEUS demonstrated more pronounced changes in responders during follow-up. Good agreement between 2D-CEUS and 3D-CEUS and MRI at 1 month.

Abbreviations: area under the curve (AUC); contrast-enhanced ultrasound (CEUS); computed tomography (CT); dynamic contrast-enhanced ultrasound (DCE-US); drug eluting beads transarterial chemoembolization (DEB-TACE); degradable starch microspheres transarterial chemoembolization (DSM-TACE); magnetic resonance imaging (MRI); transarterial chemoembolization (TACE).

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
