# Peer review of "Dynamic Contrast-Enhanced Ultrasound in the Prediction of Advanced Hepatocellular Carcinoma Response to Systemic and Locoregional Therapies"

_cancers, 2024, doi:10.3390/cancers16030551_

Round 1
Reviewer 1 Report
Comments and Suggestions for Authors
The authors want to address whether CEUS and DCE-US could be used as tools for early evaluating the treatment effect on advanced HCC. The topic is relevant in the field of using systemic or local therapy for advanced HCC.
This is a review article. It does not add anything new to the topic. However, the article could provide a comprehensive review on using DCE-US for advanced HCC.
The authors did an extensive review on this topic. To make this review a better quality, the authors can elaborate more on the relevant perspectives and possible clinical trial designs.
The conclusions are consistent with the arguments presented. The references are appropriate.
Author Response
Thanks to you kind evaluation. We further commented this point at page 12, line 371-376.
Reviewer 2 Report
Comments and Suggestions for Authors
The authors have chosen a challenging, current topic. It is a well-structured and written review, with ample and documented discussions. Although most studies evaluated small groups of patients, future clinical trials will certainly bring additional data that can argue for the introduction of dynamic contrast-enhanced ultrasound in the follow-up algorithm of patients with hepatocarcinoma during treatment.
Author Response
Dear Sirs, we would like to thank you for the suggestion: we faced this theme at pag. 12, lines 371-376.
Reviewer 3 Report
Comments and Suggestions for Authors
This review demonstrates the usefulness of DCE-US in determining the efficacy of antiangiogenetic therapies and TACE for hepatocellular carcinoma.
This paper is well organized, but should explain more about DCE-US data.
Comments
・What part of the time intensity curves does each parameter such as PE, AUS, WiR, etc. represent? Please explain in detail to Figure 1.
・Please explain with additional time intensity curves showing treatment effects. Can you add a comparison with Figure 1B?
・In Table 1, assess from the following papers:
Validation of dynamic contrast-enhanced ultrasound in predicting outcomes of antiangiogenic therapy for solid tumors: the French multicenter support for innovative and expensive techniques study.
Lassau N,et al.Invest Radiol. 2014 Dec;49(12):794-800.
・Chapter 4: It is easier to understand if a table is presented, as in Table 1.
・Page 3, last line: mean transit time (mTT) => MTT?
Author Response
Dear Sirs
We thank you for your suggestions. We modified the paper according to your note:
1) What part of the time intensity curves does each parameter such as PE, AUS, WiR, etc. represent? Please explain in detail ---> see Figure 1 and 2.
2) Please explain with additional time intensity curves showing treatment effects. Can you add a comparison with Figure 1B --> See figures 1 and 2.
3) In Table 1, assess from the following papers --> see table 1
4) Validation of dynamic contrast-enhanced ultrasound in predicting outcomes of antiangiogenic therapy for solid tumors: the French multicenter support for innovative and expensive techniques study. Lassau N,et al.Invest Radiol. 2014 Dec;49(12):794-800 --> please, see Table 1 and page 9 lines 260-277
5) Chapter 4: It is easier to understand if a table is presented, as in Table 1--> please, see Table 2
6) Page 3, last line: mean transit time (mTT) => MTT? We corrected this discrepancy in the paper.
Round 2
Reviewer 3 Report
Comments and Suggestions for Authors
The author has responded sincerely to the comments and has corrected them. I consider the journal acceptable.